# Advancing LLM Safe Alignment with Safety Representation Ranking

## Abstract

The rapid advancement of large language models (LLMs) has demonstrated milestone success in a variety of tasks, yet their potential for generating harmful content remains a significant safety concern. Existing safety guardrail approaches typically operate directly on textual responses, overlooking the rich information embedded in the model representations. In this paper, going beyond existing defenses that focus on a single safe response, we explore the potential of ranking hidden states across diverse responses to achieve safe generation. To this end, we propose Safety Representation Ranking (SRR), a listwise ranking framework that selects safe responses using hidden states from the LLM itself. SRR encodes both instructions and candidate completions using intermediate transformer representations and ranks candidates via a lightweight similarity-based scorer. Building on this framework, our approach directly leverages internal model states and supervision at the list level to capture subtle safety signals. Experiments across multiple benchmarks show that SRR significantly improves robustness to adversarial prompts, contributing a novel paradigm for LLM safety. Our code will be available upon publication.

## 1 Introduction

Recent large language models (LLMs) have achieved remarkable capabilities across a wide range of tasks. However, this power comes with serious safety and alignment concerns (Wang et al., 2024b; Ji et al., 2023; Anwar et al., 2024). Trained over massive pretrained corpora, LLMs have the potential to generate biased, toxic, or harmful content, and adversarial jailbreak prompts can coax an LLM into violating its own content guidelines (Liu et al., 2023; Wei et al., 2023a; Zou et al., 2023b). These vulnerabilities persist despite extensive alignment efforts during pre-training and post-training phases (Bai et al., 2022; Dai et al., 2024; Korbak et al., 2023). In practice, the potential for harmful outputs and the ability to bypass built-in safeguards raise significant concerns for deploying LLMs in real-world applications.

To mitigate these safety risks, prior work has explored a variety of defense mechanisms. A common strategy is decoding-time intervention, which redirects the decoding logic of the LLM during inference, through token distributions (Xu et al., 2024a; Banerjee et al., 2025) or safe prompts (Xie et al., 2023; Wei et al., 2023b; Zheng et al., 2024). For example, SafeDecoding (Xu et al., 2024a) adjusts the token distribution toward safe response distributions during decoding, while in-context defense (ICD) (Wei et al., 2023b) aligns the generation distributions to safe contexts with demonstrations. Such interventions can introduce a trade-off between safety and fluency: altering the decoding process may **degrade the model's natural performance** on benign inputs or increase inference cost. Meanwhile, post-processing-based defenses apply LLM-as-a-Judge to inspect the harmfulness of LLMs (Inan et al., 2023; Mazeika et al., 2024). Unfortunately, recent studies have shown that LLM-based safety judges are often overcautious: they **flag many benign prompts as unsafe** (so-called over-refusal) (Panda et al., 2024; Xie et al., 2025). This unreliability, *i.e.*, high false-positive rates, limits their practical use, as it can render the model unhelpful even on innocuous tasks.

In this work, we propose an alternative paradigm (which we call *Safety Representation Ranking*, **SRR**) for LLM safety that avoids alteration of the base model's generation logic and unreliable external judges. Our key idea is to generate multiple candidate responses (in parallel) to a given prompt and

then rank them by safety using the model's internal representations. This paradigm is similar to using a learned reward model to select outputs (Greve et al., 2016; Brown et al., 2024; Zhang et al., 2024a), but there exists an important twist: Traditional reward models are trained on the final generated text, often focusing on general measures of quality or alignment. In contrast, our proposed SRR explicitly targets safety by learning directly from the LLM's latent features. Therefore, existing external reward models may miss fine-grained safety cues embedded in the LLM's state vectors. Moreover, relying solely on an LLM to judge its own outputs can be unreliable and costly. By delving into the model's internal representation space, SRR can successfully detect subtle safety-critical representations (Wei et al., 2024; Zou et al., 2023a; Zheng et al., 2023) that an output-only classifier might overlook, and do so with a lightweight ranking step at inference time.

The SRR framework works in two phases. First, we identify safety-sensitive representations through contrastive training. Specifically, we construct safety contrastive groups: for each prompt, we sample examples of both safe and harmful responses, and then feed these paired responses through the LLM and extract their internal representations. Since the groups are semantically related but differ in safety, we can train a lightweight model (a single-layer Transformer) to distinguish *safe* vectors from *unsafe* ones. Through this process, SRR learns which features of the LLM's latent space correlate with safe content. Then, at inference time, we use the learned safety signals to rank candidate responses generated in parallel. In effect, SRR filters among the model's own outputs without changing how they were produced. Because it operates on the outputs after generation, SRR imposes almost no modification to the LLM's decoding logic. Its only overhead is the additional cost of scoring a few extra responses with a small model, which is negligible compared to full decoding.

We conduct comprehensive experiments to validate the effectiveness of the SRR model in identifying the safety responses across multiple datasets. Not only can SRR achieve a sufficiently high accuracy in unseen harmful prompts, but it can also generalize well across different safety evaluation datasets, demonstrating its prominent generalization ability in terms of safety ranking. Additionally, we extend our analysis in terms of other alignment perspectives like privacy and fairness, which validates the potential of SRR for diverse alignment considerations and broadens the applications of SRR.

Grounded by these empirical analyses, we characterize the practicality of SRR for serving as a safeguard module in real-world deployments. First, we incorporate SRR into LLM generation to study how it strengthens their robustness against jailbreak attacks. Additionally, we compare the natural performance of SRR with vanilla generation and other defense paradigms. Because SRR only ranks among natural outputs, the quality and correctness of benign queries remain essentially unchanged. Overall, our empirical results suggest that SRR is both a practical and effective module for LLM alignment.

Our contributions in this paper can be summarized as follows:

1. We introduce a novel paradigm, Safety Representation Ranking (SRR), which uses LLM internal representations to rank candidate responses by safety (or other alignment perspectives), without altering the model's decoding logic.

2. We demonstrate that SRR accurately selects safe outputs across diverse safety benchmarks, generalizes to novel prompts, and can be adapted to other alignment perspectives like privacy and fairness.

3. We show that integrating SRR into LLM inference significantly reduces harmful outputs under attack, with negligible impact on normal task performance.

## 2 RELATED WORK

**LLM Safe Alignment**. The issue of ensuring safe alignment in LLMs has become a longstanding challenge critical to their trustworthy deployment (Anwar et al., 2024; Ji et al., 2023; Yudkowsky, 2016). Specifically, LLMs have shown a tendency to generate harmful responses when confronted with malicious requests. While current alignment techniques have improved at mitigating these risks to some extent, they still tend to be superficial and inadequate (Qi et al., 2024). Additionally, inference-time defenses can reduce the success rate of these attacks, but they often struggle with a significant drawback of rejecting benign inputs, leading to over-refusal issues. The underlying mechanism of such issues is that these distribution-based or prompt-based defenses commonly change

the decoding strategies of LLMs, making their generation distributions favor refusals. Thus, ensuring safe alignment whilst maintaining the generation distribution stands for a viable solution for these risks.

**Safety Representations of LLMs**. Building on the representation engineering techniques of LLMs (Zou et al., 2023a; Zhang et al., 2024c), which examine LLM dynamics through the lens of hidden space with perspective-specific data, recent research has revealed the existence of safety representations within these models (Wei et al., 2024; Zheng et al., 2024). Specifically, low-dimensional and structured representations emerge in the hidden states of LLMs, which indicate their safety status. When these representations are activated in specific directions, the LLM can successfully recognize and refuse harmful prompts that go against its ethical guidelines. Conversely, when the activations move in the opposite directions, the LLMs fail to reject harmful inputs and display jailbreak behavior. This interesting property has attracted significant research interest aimed at locating and interpreting these representations (Chen et al., 2024; Zhao et al., 2025). Nonetheless, effective methods for leveraging them to enhance the safety of LLMs remain underexplored.

**Ranking-based LLM generation**. A variety of rule-based generation methods have been proposed to improve language model performance, including top-$k$ sampling (Fan et al., 2018; Holtzman et al., 2018), temperature-based sampling (Ficler & Goldberg, 2017), and nucleus sampling (Holtzman et al., 2020). Beyond these, more refined algorithms have been developed to focus on specific tasks. For example, Wang et al. (2023); Wang & Zhou (2024) leverage majority voting to improve LLM reasoning. Xu et al. (2024b); Li et al. (2023); Zhang et al. (2024d) employ carefully designed decoding methods to generate responses that better align with specific requirements in constrained scenarios. Recent studies (Setlur et al., 2024; Wang et al., 2024a; Zhang et al., 2024b; Trung et al., 2024) train additional reward models, scaled even equivalently to the base models, to perform reranking for specific tasks. However, these approaches are either rule-based, task-specific, or impose significant computational overhead, inherently limiting their performance potential and application scope. To overcome these limitations, we propose a more general and lightweight ranker to optimize inference-time computation specialized for safe alignment.

## 3 METHODOLOGY

In this section, we present our proposed Safety Representation Ranking (SRR), a listwise learning-to-rank framework for scoring LLM responses by safety. Given an instruction, SRR generates a set of candidate completions and ranks them such that safe responses receive higher scores than unsafe ones. The core idea is to extract internal representations from a frozen base LLM and train a lightweight transformer ranker to assess instruction-response compatibility. Below, we describe the key components of SRR: candidate response generation, ranker architecture, and optimization with a listwise ranking objective.

### 3.1 CANDIDATE RESPONSE GENERATION

To construct candidate lists for training, we sample the base LLM multiple times using stochastic decoding with moderate temperature. This yields a diverse set of $m$ plausible responses $\{resp_1, \ldots, resp_m\}$. for each instruction. We remove duplicates and include both benign and adversarial candidates by injecting jailbreak prompts (Wei et al., 2023b; Zou et al., 2023b). This helps ensure that the candidate pool contains both safe answers and hard negatives (unsafe answers) for training. Each response is labeled with a binary safety tag $y_i \in \{0, 1\}$, where $y_i = 1$ indicates a safe response. For training, we construct tuples of the form $(inst, \{resp_i, y_i\}_{i=1}^m)$, where each list includes at least one safe and one unsafe response.

### 3.2 RANKER MODEL ARCHITECTURE

The core of SRR is a neural ranker that computes a compatibility score between an instruction and each candidate response. We build this ranker as follows:

- **Step 1. Representation extraction:** We use the base LLM as a fixed feature extractor. For each textual input (instruction or response), we run it through the LLM and take the hidden-state vector at a selected layer as its representation. Concretely, let $\mathbf{h}_{\text{inst}} \in \mathbb{R}^d$ be the

hidden vector for the instruction (the state of the last token in the sequence) at the chosen layer, and let $\mathbf{h}_{\mathrm{resp},i} \in \mathbb{R}^d$ be the hidden vector for the $i$-th response. Since the backbone is trained for next-token prediction, the final layers tend to overfit to this specific task. In contrast, intermediate layers typically provide more comprehensive representations of the preceding context, making them better suited for capturing the overall features required for ranking (Skean et al., 2024). Therefore, we adopt intermeidiate layers to capture high-quality semantic content.

- **Step 2. Transformer encoder:** We map each high-dimensional LLM vector (typically $d = 4096$) to a lower-dimensional space using a shared learned linear projection. This makes the downstream transformer encoder more lightweight and efficient. We concatenate the projected vectors into a sequence:

$$[\mathbf{h}_{\mathrm{inst}}, \mathbf{h}_{\mathrm{resp},1}, \ldots, \mathbf{h}_{\mathrm{resp},m}]. \tag{1}$$

This sequence is then passed through a Transformer encoder (single-layer in our implementation). The Transformer's self-attention layers let the instruction embedding interact with each response embedding. After passing through the encoder, we obtain output vectors $\mathbf{o}_{\mathrm{inst}}$ and $\mathbf{o}_{\mathrm{resp},i}$ corresponding to the instruction and each response, respectively. Intuitively, $\mathbf{o}_{\mathrm{inst}}$ is the contextualized instruction representation (having attended to all responses) and $\mathbf{o}_{\mathrm{resp},i}$ is the $i$th response representation attended to the instruction.

- **Step 3. Similarity computation:** From these encoder outputs we compute a similarity score $s_i$ for each response. We use cosine similarity:

$$s_i = \cos(\mathbf{o}_{\mathrm{inst}}, \mathbf{o}_{\mathrm{resp},i}) = \frac{\mathbf{o}_{\mathrm{inst}}^{\top}\mathbf{o}_{\mathrm{resp},i}}{\|\mathbf{o}_{\mathrm{inst}}\|\|\mathbf{o}_{\mathrm{resp},i}\|}. \tag{2}$$

These scores $s_i \in [-1, 1]$ measure the alignment between instruction and responses in the embedding space, which are used as unnormalized logits for ranking, with a temperature scaling parameter $\tau$ applied before softmax to control sharpness.

### 3.3 TRAINING OBJECTIVES AND OVERALL PIPELINE

We train the ranker end-to-end (keeping the base LLM frozen) using a listwise ranking loss. For safe *v.s.* unsafe contrastive training, we interpret the similarity scores $s_i$ for a list of $m$ candidates as unnormalized logit scores. We then compute a softmax probability for each response:

$$\hat{p}_i = \frac{\exp(s_i/\tau)}{\sum_{j=1}^{m} \exp(s_j/\tau)}. \tag{3}$$

For the ranking labels, we also define a ground-truth probability distribution $p^*$ over the list, which places all mass on the safe responses. For instance, if there are $k$ safe responses among the $m$, we set $p_i^* = 1/k$ for each safe response with $y_i = 1$ and 0 for unsafe ones with $y_i = 0$. Then we minimize the Kullback–Leibler divergence:

$$\mathbb{D}_{\mathrm{KL}}\left(p^* \,\|\, \hat{p}_i\right) = \sum_{i=1}^{m} p_i^* \log \frac{p_i^*}{\hat{p}_i}. \tag{4}$$

This loss, as a standard choice (Purpura et al., 2022; Liu et al., 2024), encourages the model to assign high probability to safe candidates. In effect, the ranker is trained so that the instruction and safe responses have higher cosine similarity than instruction-unsafe pairs.

### 3.4 SAFETY RANKING DURING INFERENCE

Given the training pipeline above, SRR learns to map instructions and responses into a joint embedding space where safety alignment is captured by similarity. In the inference stage, for any new prompt $q$ and its candidate outputs, the ranker can compute similarity scores and produce a safety-based ranking without further supervision, then return the highest safety-ranked response. A potential concern is that all generated responses are unsafe given one harmful prompt, then the ranking mechanism

---

**Algorithm 1** Safety Representation Ranking (SRR)

---

**Require:** Instruction in training data, LLM $f$, response generator $\mathcal{G}$, ranker $g_\theta$, temperature $\tau$

 **Training Phase:**
1: **for** each instruction in training data **do**
2:     $\{\text{resp}_1, \ldots, \text{resp}_m\} \leftarrow \mathcal{G}(\text{instruction})$     ▷ Generate diverse candidate responses
3:     $y_i \leftarrow$ safety label for each $\text{resp}_i$     ▷ 1 for safe, 0 for unsafe
4:     $\mathbf{h}_{\text{inst}} \leftarrow f(\text{inst})$, $\mathbf{h}_{\text{resp},i} \leftarrow f(\text{resp}_i)$ for $i = 1 \ldots m$     ▷ Extract LLM features
5:     $[\mathbf{o}_{\text{inst}}, \mathbf{o}_{\text{resp},1}, \ldots] \leftarrow g_\theta([\mathbf{h}_{\text{inst}}, \mathbf{h}_{\text{resp},1}, \ldots])$     ▷ Transformer-based contextual encoding
6:     $s_i \leftarrow \cos(\mathbf{o}_{\text{inst}}, \mathbf{o}_{\text{resp},i})$     ▷ Compute cosine similarity score
7:     $\hat{p}_i \leftarrow \dfrac{\exp(s_i/\tau)}{\sum_j \exp(s_j/\tau)}$     ▷ Normalize scores via softmax
8:     $p_i^* \leftarrow \dfrac{1}{k}$ if $y_i = 1$, else 0     ▷ Uniform probability on $k$ safe responses
9:     $\mathcal{L} \leftarrow \text{KL}(p^* \| \hat{p})$     ▷ Listwise loss
10:     Update $\theta$ to minimize $\mathcal{L}$

 **Inference Phase:**
11: Given a new instruction $q$, generate candidate $\{\text{resp}_1, \ldots, \text{resp}_m\}$ (in parallel)
12: Repeat steps 2-6 for the generated responses to compute $s_i$
13: **return** Responses ranked by descending $s_i$

---

may be ineffective. In practice, we slightly modify the generation hyperparameters, like decoding temperature, to ensure the generation is diverse enough to include at least one safe response in all responses (more details in the next section).

Overall, a pseudo-algorithm is provided in Algorithm 1. First, SRR generates diverse candidate responses for each instruction during the training phase (line 2). It then extracts features from the LLM and uses a transformer-based ranker to compute similarity scores between instructions and responses (line 3-6). These scores are normalized via softmax and compared to ground-truth probabilities to compute a listwise loss, which is used to update the ranker (line 7-10). During inference, the algorithm repeats the feature extraction and similarity computation steps to rank responses based on safety.

## 4 EVALUATION

In this section, we conduct comprehensive evaluations to show the effectiveness of SRR across diverse alignment perspectives, including safety, privacy, and fairness, starting with the overall setup. To further demonstrate the generality of our approach, we also evaluate its generalization ability on other datasets. We finally present that the natural performance in math and coding does not deteriorate after attaching the ranker to the model.

### 4.1 EXPERIMENT SET-UP

**Models and datasets.** In our experiment, we apply three popular LLMs, including (1) **Qwen2.5-7b-Instruct** (Yang et al., 2024) (2) **Mistral-7-v0.3** (Jiang et al., 2023), and (3) **Vicuna-7b-v1.5** (Zheng et al., 2023). For datasets, we apply **Harmbench** (Mazeika et al., 2024), **SorryBench** (Xie et al., 2025), and **JailbreakBench** (Chao et al., 2024). The HarmBench dataset here refers to the standard section of the Harmbench dataset, which includes 200 different harmful prompts in various areas. The SorryBench and the JailbreakBench have similar contents. For each dataset, we extract 50 of them as the training dataset, and the rest is used as the testing dataset. For each prompt in a dataset, we sample answers from the base model using In-context Attack and In-context Defense, each 20 times. After sampling, we check whether the answers are valid by examining the keywords in them. For safe answers, we examine "Sorry", "unable", "illegal", and "understand". For harmful answers, we examine "sure", "certainly". We then filter the answers using the above criterion to get high-quality data.

Table 1: Ranking accuracy of SRR in distinguishing safe and harmful prompts.

| Source Dataset | Method | Qwen | Model Mistral | Vicuna | **Average** |
|---|---|---|---|---|---|
| Harmbench | Baseline | 41.18 | 35.21 | 57.60 | 44.66 |
| | Ours | **82.35** | **91.55** | **90.40** | **88.10** |
| SorryBench | Baseline | 56.72 | 52.82 | 55.26 | 54.93 |
| | Ours | **85.57** | **90.15** | **87.98** | **87.90** |
| JailbreakBench | Baseline | 70.00 | 67.39 | 50.00 | 62.46 |
| | Ours | **80.00** | **95.65** | **95.24** | **90.30** |

**Metrics**. In the experiments in this section, we ask the model to choose between the safe and harmful answers, verifying its correctness by referring to the ground-truth label from the data generation process for safe/unsafe data.

**Ranker Settings**. In all experiments, the rankers are implemented using a single Transformer block. The trainable parameters of the ranker model is less than 5M. They operate on features extracted from approximately the bottom 25% of the base model's layers. During training and evaluation, every data group includes two candidate answers. The ranker is trained to distinguish the answers as safe and harmful. The hyperparameters are set as follows: learning rate is set to 0.001, weight decay is set to 0.0001, dropout is set to 0.1, and momentum is set to 1.0.

**Baseline**. The baseline of the experiment adopts a reward model to rate answers generated by the base model. A pretrained GPT2 (Radford et al., 2019) is used as the reward model in the experiment. Small as it seems, a GPT2 model is still 20 times larger than the ranker model.

### 4.2 OVERALL EVALUATION

We use the transformer-architected ranker to improve the safety of different models on different datasets. As depicted in Table 1, our method significantly outperforms the reward model in all base models and datasets. The accuracy of many experiments reaches 90%. Our lightweight method significantly outperforms the reward model (gpt2), despite being far smaller in scale. Specifically, when Qwen is used as the base model, the ranker reaches 82.35%, 91.55%, 90.40% respectively on three datasets. Similarly, the results are 85.57%, 90.15%, 87.98% when the base model is Mistral. Finally, the performance is 80.00%, 95.65%, 95.24% when the base model is Vicuna. This implies that rankers can adapt to even larger models.

### 4.3 CROSS DATASET VALIDATION

To further evaluate the generalization capability of our SRR framework across different safety benchmarks, we conduct cross-dataset validation experiments. We apply the ranker trained on one dataset to other unseen datasets. This experimental setup helps us demonstrate whether the model can effectively identify and prioritize safe responses regardless of the dataset's specific characteristics or the types of adversarial prompts it contains.

The results in Table 2 show that our SRR framework achieves consistently strong cross-dataset performance across all three LLMs. When trained on one dataset and evaluated on another, SRR maintains a high level of accuracy in distinguishing safe from harmful responses. For instance, a ranker trained on Harmbench achieves 77.02% average accuracy on SorryBench and 86.40% on JailbreakBench. Similarly, a ranker trained on SorryBench achieves 82.20% on Harmbench and 81.03% on JailbreakBench. This cross-dataset effectiveness demonstrates that SRR's safety signal is not overly specialized to any particular dataset but instead captures generalizable features of safety within the LLM's internal representations.

This ability to generalize across different safety benchmarks is crucial for real-world deployment. In practical applications, LLMs may encounter a wide variety of adversarial prompts that differ significantly from those seen during training. The strong cross-dataset performance of SRR suggests that it can serve as a robust safeguard module, effectively filtering out harmful responses even when

Table 2: Cross-dataset ranking accuracy of SRR in distinguishing safe and harmful prompts.

| Source Dataset | Evaluation Dataset | Model | | | |
| | | Qwen | Mistral | Vicuna | **Average** |
| --- | --- | --- | --- | --- | --- |
| Harmbench | SorryBench | 76.96 | 88.06 | 66.04 | 77.02 |
| | JailbreakBench | 80.00 | 93.48 | 85.71 | 86.40 |
| SorryBench | Harmbench | 76.47 | 90.14 | 80.00 | 82.20 |
| | JailbreakBench | 77.78 | 89.13 | 76.19 | 81.03 |
| JailbreakBench | HarmBench | 79.41 | 89.44 | 90.40 | 86.42 |
| | SorryBench | 72.41 | 87.16 | 78.59 | 79.39 |

Table 3: Ranking accuracy of SRR in distinguishing infringement and benign inputs.

| Dataset | Model | | | |
| | Qwen | Mistral | Vicuna | **Average** |
| --- | --- | --- | --- | --- |
| Harmcopy | 98.08 | 95.83 | 89.74 | 94.28 |

the specific types of attacks vary. This provides evidence that SRR's approach of leveraging internal model representations for safety ranking is both versatile and adaptable to diverse safety challenges.

### 4.4 EXTENSION TO OTHER ALIGNMENT PERSPECTIVES

In this part, we also extend the application of our Safety Representation Ranking (SRR) framework to other critical alignment perspectives beyond general safety, namely privacy and fairness. These dimensions are essential for ensuring that LLMs not only avoid harmful content but also respect user privacy and produce unbiased, equitable responses. Evaluating SRR's effectiveness in these areas helps to demonstrate its versatility and potential for broader alignment applications.

**Privacy**. To evaluate the potential of SRR in addressing privacy concerns, we conducted experiments on the Harmcopy dataset (Mazeika et al., 2024), which contains prompts related to privacy infringement. The results are presented in Table 3, showing that SRR achieves a high accuracy rate in distinguishing between privacy-infringing and benign prompts across all models. In particular, Qwen demonstrates the highest accuracy of 98.08%, followed by Mistral with 95.83% and Vicuna with 89.74%. The average accuracy across all models is 94.28%, indicating that SRR is effective in identifying privacy-related safety concerns.

**Fairness**. To assess the effectiveness of SRR in ensuring fairness, we conducted experiments on the BBQ dataset (Parrish et al., 2021). This dataset is designed to evaluate the model's ability to avoid generating responses that may contain biases or unfair content. The results are presented in Table 4, indicating that SRR achieves better accuracy in identifying and mitigating biased or unfair responses. Note that this dataset is a three-category classification problem, and baseline performance is even less than 33% three categories (Parrish et al., 2021), thus SRR indeed improves this result in a large margin.

Overall, the results demonstrate the initial potential of SRR in addressing privacy and fairness concerns for future advancement in this critical area of LLM alignment. The strong performance in the privacy and fairness context further validates the generalizability of SRR, whose ability to adapt to privacy or fairness-specific prompts shows that SRR can capture fine-grained safety signals related to different alignment perspectives beyond just general harmful content. This makes it a versatile and efficient solution for enhancing the privacy safeguards in LLM applications.

### 4.5 BRIEF SUMMARY

In this section, we have comprehensively evaluated the effectiveness of our proposed SRR framework across various dimensions of LLM safety and alignment. Our experiments demonstrate that SRR achieves significant improvements in identifying and prioritizing safe responses over harmful ones,

Table 4: Ranking accuracy of SRR in distinguishing unfair and benign inputs.

| Dataset | Model | | | Average |
| --- | --- | --- | --- | --- |
| | Qwen | Mistral | Vicuna | |
| BiasedBenchmark for QA (BBQ) | 54.82 | 52.09 | 50.64 | 52.52 |

Table 5: Real-world ranking accuracy of different methods in distinguishing safe and harmful prompts across HarmBench, JailbreakingBench, and SorryBench.

| Method | HarmBench | | | JailbreakingBench | | | SorryBench | | |
| --- | --- | --- | --- | --- | --- | --- | --- | --- | --- |
| | Qwen | Mistral | Average | Qwen | Mistral | Average | Qwen | Mistral | Average |
| First | 82.52 | 54.43 | 68.48 | 16.25 | 32.91 | 24.58 | 84.28 | 46.22 | 65.25 |
| SRR/Ranker | **83.22** | **63.29** | **73.26** | **38.75** | **39.24** | **39.00** | **86.16** | **67.23** | **76.70** |

with high accuracy across multiple safety benchmarks. The cross-dataset validation further confirms the generalizability of SRR, showing its ability to adapt to different types of adversarial prompts and datasets without being overly specialized. Additionally, our extension to privacy-related prompts reveals SRR's potential in mitigating privacy-infringing outputs, achieving a strong accuracy rate. Even in the context of fairness, where the task is more nuanced, SRR shows a foundational capability to distinguish between biased and unbiased responses, though with moderate accuracy that suggests room for further enhancement. Overall, these results highlight SRR's versatility and effectiveness as a safeguard module that can be integrated into LLM inference to significantly reduce harmful outputs under attacks.

## 5 DISCUSSION

This section further discusses the considerations for SRR in practical deployment. We focus on two fundamental research questions (RQs):

**RQ1**. To what extent can SRR mitigate safety alignment issues?

**RQ2**. How does SRR impact the natural performance of LLMs?

### 5.1 RQ1: REAL-WORLD APPLICATION

Recall that we mainly apply the classification accuracy as the main metric to evaluate the precision of SRR in ranking the safety of multiple responses. In this part, we further explore how SRR can improve the safety alignment of LLMs, since aligned LLMs have already exhibited certain robustness against harmful prompts. To this end, we incorporate SRR during real-time inference of the protected LLMs, rather than classifying simulated harmful or safe responses. We also consider practical jailbreak attacks to demonstrate the robustness of SRR. The baseline in this experiment is "first accuracy", which means choosing the answer with the highest possibility generated by the base model.

The results shown in Table 5 demonstrate that SRR significantly enhances the safety alignment of LLMs in real-world applications. When integrated into the inference process of protected LLMs, SRR demonstrates robust performance against practical jailbreak attacks. This indicates that SRR can effectively improve the safety mechanisms of LLMs, reducing their vulnerability to adversarial prompts. By leveraging the model's internal representations, SRR provides an efficient and effective safeguard without compromising the natural performance of the LLMs. Overall, these findings support the practical utility of SRR as a valuable tool for improving the safety and reliability of LLMs in real-world scenarios.

### 5.2 RQ2: NATURAL PERFORMANCE

As discussed in earlier sections, a key advantage of SRR is that it does not intervene in the decoding process of the base language model. This allows SRR to be seamlessly applied at inference time

Table 6: Accuracy (%) comparison on the MATH and MBPP dataset when responses are ranked using SRR trained on different safety datasets. Natural (w/o SRR) denotes baseline performance without SRR defense mechanism.

| Source Dataset | Natural (w/o SRR) | HarmBench | SorryBench | JailbreakBench |
|:---:|:---:|:---:|:---:|:---:|
| MATH | 68.7 | 69.1 | 68.5 | 68.6 |
| MBPP | 60.6 | 61.6 | 61.4 | 60.8 |

without modifying generation behavior, thereby preserving the model's natural task performance. In this section, we empirically validate this claim using a mathematical reasoning benchmark. We evaluate SRR using the MATH dataset (Hendrycks et al., 2021) for mathematical problems and the MBPP dataset (Austin et al., 2021) for coding problems. MATH contains 12,500 competition-level math problems spanning seven topics and five difficulty levels, and the MBPP dataset is a popular benchmark for code generation, comprising 500 basic Python programming tasks to evaluate the ability of models to generate functional code from text. To assess performance, we extract the final answer from each model-generated response and compare it against the ground-truth answer. We use Qwen2.5-7B-Instruct as the base model. For each instruction, we sample 10 completions and apply the SRR ranker, which is trained solely on safety datasets, to rank them by their predicted safety. The top-ranked response is selected as the final answer. We then compare the answer accuracy of the ranked responses against the accuracy obtained by the base model's default outputs.

The results are shown in Table 6. Across all settings, the accuracy of the SRR-ranked completions remains nearly identical to the base model's natural accuracy (68.7%). In fact, slight fluctuations ($\pm0.2\%$) are observed depending on which safety dataset the ranker was trained on, but these differences fall within the margin of noise and do not indicate degradation in performance. Notably, this result holds despite the SRR ranker being trained exclusively on safety supervision signals, without any exposure to mathematical reasoning data. This demonstrates that the SRR scoring mechanism does not introduce unintended bias toward specific task domains or alter the correctness of model outputs in benign settings.

## 6 LIMITATIONS

While SRR demonstrates strong performance in LLM safety alignment, several critical limitations warrant further investigation from a practical deployment and methodological robustness perspective. First, SRR relies on extracting features from the intermediate layer representations, but it lacks a systematic analysis of how layer selection impacts ranking accuracy. This heuristic layer choice lacks quantitative validation, which may undermine the method's stability when applied to LLMs with different architectural configurations. Second, the framework's safety judgment is tied to binary labels (safe/unsafe) and keyword-based validation, but its current form fails to address fine-grained safety risks (e.g., subtly biased content). Third, though SRR resists conventional jailbreak prompts, it has not been validated against adaptive attacks targeting its core similarity.

## 7 CONCLUSION

In this paper, we introduced Safety Representation Ranking (SRR), a novel listwise ranking framework that leverages the internal representations of LLMs to select safe responses without altering the model's decoding logic. Through contrastive training, SRR identifies safety-sensitive features within the LLM's hidden states and uses them to rank candidate responses based on safety. Our method not only improves robustness against adversarial prompts but also generalizes well across different safety evaluation datasets. Furthermore, SRR demonstrates potential for addressing other alignment perspectives such as privacy and fairness. Experimental results indicate that SRR significantly reduces harmful outputs under attack while maintaining performance on benign tasks. Overall, SRR serves as a practical and effective safeguard module for LLM alignment, offering a new paradigm for enhancing the safety and reliability of LLMs in real-world applications.

## REPRODUCIBILITY STATEMENT

Our code will be available upon publication. All models and datasets can be accessed through `huggingface.co`.

## ETHICS STATEMENT.

This work complies with the ICLR Code of Ethics. While our methods are designed to generate safe contents, they may be applied in contexts with societal implications, including risks related to bias, fairness, and privacy. We encourage responsible use and declare no conflicts of interest.

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

## A   THE USE OF LARGE LANGUAGE MODELS (LLMS)

In this work, LLMs are primarily employed for polishing the language of the manuscript to ensure grammatical correctness and coherence. Importantly, all conceptual development, experimental design, and result interpretation are conducted independently by the authors. The use of LLMs is strictly limited to auxiliary tasks, ensuring that the scientific contributions of this paper remain entirely unaffected by such tools.

