# OpenReview forum: "Advancing LLM Safe Alignment with Safety Representation Ranking"
_ICLR.cc/2026/Conference — Submitted to ICLR 2026_

### Official Review · Reviewer_2roF · 2025-10-26

**Soundness:** 1
**Presentation:** 2
**Contribution:** 2
**Rating:** 2
**Confidence:** 4

**Summary:**

The paper proposes Safety Representation Ranking (SRR): generate multiple candidate completions from a frozen base LLM, extract intermediate hidden states for the instruction and each candidate, pass them through a small one‑layer Transformer ranker, compute cosine‑similarity scores, and select the “safest” response. The authors evaluate on HarmBench, SorryBench, and JailbreakBench (train on 50 prompts per dataset; the rest test), show higher “accuracy” than a GPT‑2 reward‑model baseline at picking the safe item, claim gains under “practical jailbreak attacks,” and report negligible impact on MATH/MBPP accuracy. Code is promised upon publication.

**Strengths:**

(1) Clear, lightweight idea: Using frozen internal states plus a tiny ranker is simple and potentially deployable; the method is well summarized in Algorithm 1.

(2) Cross‑dataset signal: Reported cross‑dataset results (e.g., train on HarmBench → test on SorryBench/JailbreakBench) suggest the ranker learns some generalizable safety cues.

(3) Separation from decoding: Because SRR chooses among naturally generated candidates and does not alter token distributions, it targets safety without the over‑refusal failure mode often associated with decoding‑time interventions (a reasonable design goal).

**Weaknesses:**

(1) Baselines are far too weak- The sole baseline is a small GPT‑2 reward model; this does not represent today’s de facto defenses.

(2) Over‑refusal evaluation is missing- The paper argues SRR preserves helpfulness on benign inputs but does not evaluate on standard over‑refusal benchmarks.

(3) Safety claims without strong jailbreaks- The “real‑world” section mentions “practical jailbreak attacks” but does not name or systematically evaluate against high‑pressure, black‑box jailbreakers. This makes the robustness claims very weak.

(4) Comparability to the base model is unclear- Most main tables compare SRR to a reward‑model baseline rather than reporting absolute ASR of the base model itself. The reported metric is “accuracy”, not attack success or harm rate. ASR is more suitable since this is a generation task and not a classification one.

(5) Ablations and analysis are limited. The method’s performance depends on which layer provides the features, yet no systematic layer-wise analysis or sweep is conducted. This aspect is crucial for assessing the practicality and general applicability of the approach.

**Questions:**

(1) Could the authors justify their choice of a GPT-2 reward model as the only baseline?
Specifically, why were stronger and more representative defenses—such as LLM-as-Judge systems (e.g., Llama Guard), SafeDecoding, or recent representation-editing approaches like [1]—omitted from comparison?

(2) How does SRR perform on standard over-refusal or false-refusal benchmarks (e.g., XSTest, OR-Bench, PHTest, FalseReject)?
Since the paper claims SRR preserves helpfulness on benign inputs, please provide quantitative evidence that it does not over-reject benign queries.

(3) Which specific jailbreak attacks were used in the “real-world” evaluation of Table 5?
Have the authors tested SRR under high-pressure, black-box jailbreaks such as PAIR, PAP, or AutoDAN-Turbo, and if not, how can the reported robustness claims be substantiated?

(4) Could the authors report absolute attack-success rates (ASR) and harmfulness scores for the base models themselves?
Why is “accuracy” used instead of ASR or harm-rate metrics, which are more appropriate for generation tasks, and how should readers interpret the reported numbers in relation to real-world safety performance?

(5) The limitations note that SRR’s layer selection is heuristic. Please perform or at least report preliminary results from a systematic layer-wise sweep to substantiate how layer choice impacts SRR’s robustness and transferability.

[1] Wang, Xinpeng, et al. "Surgical, cheap, and flexible: Mitigating false refusal in language models via single vector ablation." arXiv preprint arXiv:2410.03415 (2024).

---

### Official Review · Reviewer_q4Xz · 2025-10-30

**Soundness:** 2
**Presentation:** 2
**Contribution:** 2
**Rating:** 2
**Confidence:** 4

**Summary:**

This work proposes Safety Representation Ranking (SRR), which utilizes hidden representations to rank multiple candidate responses by their safety. The Authors utilize a transformer encoder trained on instruction and responses representations to rank the responses based on the similarity score between encoded response candidates and the instruction. To evaluate their method, the Authors utilize 3 open-source models and 3 harmfulness-related datasets. As a baseline, the Authors utilize a pretrained GPT-2 as a reward model and compare ranking accuracy on Harmbench, SorryBench, and JailbreakBench. Additionally, the Authors show how SRR performs on BBQ, MATH, and MBPP datasets to highlight their

**Strengths:**

The Authors introduce an alternative to safe generation by using a safety reranker with multiple response candidates. The SRR is validated on multiple datasets and models. The Authors have demonstrated how SRR performs on benign datasets and bias datasets. The experiments in the article are insufficient to ensure that the SRR would be beneficial in a real-life scenario.

**Weaknesses:**

* The Authors don’t explain in detail how $h_{resp,i}$ is calculated, as the hidden representations of responses have an additional dimension of length in terms of tokens.
* In L163-164, the Authors state: “Since the backbone is trained for next-token prediction, the final layers tend to overfit to this specific task,” but don’t provide any evidence/literature for this statement.
* I see no theoretical or intuitive reason why the usage of a transformer encoder would be beneficial for this encoding part. I believe that using a simple projection could be enough to ensure that safety-related features are extracted from the instruction and responses.
* The SRR only uses this safety-related ranking, but I believe that not testing how it influences the text quality completely undermines the utility of this method. I imagine that this method would always score highly for a simple refusal message. To address this issue, Authors should utilize a benign dataset such as AlpacaEval[1] to ensure that the quality of the text does not suffer from SRR. Currently, SRR is evaluated on how effective it is in classifying safe and unsafe responses.
* In L303-304, the Authors state: “This implies that rankers can adapt to even larger models.”, but all the models used in experiments have 7B params, which doesn’t allow for such a strong statement.

[1] AlpacaEval: An Automatic Evaluator of Instruction-following Models, Xuechen Li and Tianyi Zhang and Yann Dubois and Rohan Taori and Ishaan Gulrajani and Carlos Guestrin and Percy Liang and Tatsunori B. Hashimoto, 2023

**Questions:**

* Can the Authors compare SRR to using other safety classifiers, such as Llama-Guard, on generated responses to show the advantage of SRR over using other safety classifiers for ranking the candidates?
* Does Natural in Table 6 also utilize more than one generation? To make this comparison fair, I think that it is necessary to use at least a beam search for the Natural generation.
* Can the Authors provide any evidence that the usage of the transformer encoder over a non-sequence encoder, such as MLP, is beneficial?

---

### Official Review · Reviewer_i24y · 2025-10-31

**Soundness:** 2
**Presentation:** 2
**Contribution:** 1
**Rating:** 2
**Confidence:** 4

**Summary:**

This paper studies the safety of Large Language Models (LLM) generation and attempts to address it. It proposes a simple hook that ranks multiple generations of the LLM at a given received input prompt, and outputs the one that is the safest. The proposed approach is simple, and the experiments show potential positive performance gain.

**Strengths:**

The main strengths of this paper are:

1) The problem this paper studies is important: Aligning LLM's generation to be more safe.

2) The proposed approach is simple and easy to implement.

3) The experiments span three different LLM architectures, and a few datasets.

**Weaknesses:**

Despite the paper's strengths, there are major weaknesses that need to be addressed before getting this paper accepted:

1) From the methodology side, I have the following two criticisms that need to be addressed:

1a) Leveraging the embedding of the last generated token as an embedding for the entire instruction/response seems incorrect and has too little information about the entire sequence. One should employ the embeddings for all instruction/response tokens (perhaps average them).

1b) The definition of $s_i$ as a logit for safety is not convincing nor motivated. Why would the generation be safe if the input and output embeddings are correlated?

2) The experiments in this work, despite spanning multiple models and datasets, are questionable for the following reasons:

2a) Let us start from the construction of the datasets: an answer having one of these keywords (Sorry, unable, ...) does not imply safety. One can construct many counter examples to this setting making both the training and testing sets questionable.

2b) The use of GPT-2 as a ranker for this safety based evaluation is incorrect. There is a wealth of literature developing guard models that can be used as safety evaluator for both training and evaluation (such as Llama-Guard, Granite Guardian, Shield Gemma, ...).

2c) Comparing the developed method against the naive LLM as a baseline is unfair. One should compare the developed method against stronger baselines such as prompt tuning, Low rank adaptation, ...etc.

2d) Many of the experimental setups are unclear. For example, what is "First" in Table 5, and how does it compare to the baseline performance in Table 1.

3) The writing of this paper can be vastly improved. For example, all figure captions are not very informative as they do not include the experimental setting, nor baseline descriptions. Here are a few suggestions to improve the readability of this paper:

3a) Include a pipeline figure describing the proposed method

3b) Improve Table captions to include more information about the experiment setting for the numbers reported in that table.

3c) Provide an equation for the reported metric in the experiments.

**Questions:**

Please refer to the weaknesses section.

---

### Official Review · Reviewer_ZLF4 · 2025-11-01

**Soundness:** 2
**Presentation:** 3
**Contribution:** 2
**Rating:** 2
**Confidence:** 4

**Summary:**

This paper introduces Safety Representation Ranking (SRR), a framework designed to enhance the safety alignment of LLMs during inference. The core premise of SRR is to leverage the intermediate-layer representations of LLMs to rank multiple candidate responses for a given prompt based on their safety. The method involves generating candidate completions through sampling and extracting their hidden state vectors from a selected intermediate layer of the LLM, and a lightweight, trainable Transformer-based ranking models trained with a KL-divergence based ranking objective. During inference, the ranking model can rate different responses of the LLM from an instruction, and therefore provide a safer response. The paper evaluates SRR across three base LLMs and three safety benchmarks demonstrating high ranking accuracy in distinguishing safe from harmful responses compared with the baseline. Experiments also show that the SRR generalizes effectively in cross-dataset evaluations and  other alignment perspectives like privacy and fairness.

**Strengths:**

1. This paper introduces a lightweight inference-time aligning method for LLMs, which does not need heavy training. While ranking responses itself is not new, using intermediate hidden states is an underexplored approach.
2. The paper is generally well-written, and the methodology is well-structured and described in detail.

**Weaknesses:**

1. **The motivation of the proposed method is inadequate, which also limits the novelty of the paper.** The paper's central claim that ranking candidate responses using internal representations is superior to using final text is not sufficiently motivated or proven. It states that traditional reward models "may miss fine-grained safety cues embedded in the LLM’s state vectors," but provide no empirical evidence to support this critical claim. The entire motivation for using intermediate representations hinges on this unverified hypothesis. A direct and necessary ablation is missing: comparing the performance of SRR using internal states against another ranking model training on the same dataset but using only the final text embeddings of the responses. Without this comparison, it is impossible to conclude that the intermediate representations are the key factor for the observed performance, as opposed to, for instance, the listwise ranking objective itself or the specific architecture of the ranker.

2. **The experimental settings, including the metric and baseline are not convincing enough.** Firstly, the primary metric, "ranking accuracy," is an indirect measure compared to the safety rate or attack success rate under jailbreak scenarios, which is widely used for LLM safety research. A high ranking accuracy does not directly translate to a model that robustly refuses to answer harmful instructions in practice. Secondly, the choice of baselines is limited and fails to contextualize SRR within the current research landscape. Comparing solely against a GPT-2 based reward model is inadequate. To demonstrate its contribution, SRR must be compared against contemporary and strong inference-time defense methods such as ARGS [1] and SafeDecoding [2]. The absence of these comparisons leaves the reader uncertain of whether SRR offers a tangible improvement over existing, and potentially simpler, techniques.


3. **The analysis is not sufficient enough to demonstrate the advantages of SRR.** The paper lacks a thorough analysis of SRR, including (1) Computational Overhead: While the ranking model itself is small, the overall inference cost of SRR includes generating multiple candidate responses. The total computational overhead of this process, and how it compares to the cost of other defense mechanisms (e.g. SafeDecoding), is not analyzed. (2) Layer Selection Rationale: The decision to use representations from "the bottom 25% of the base model's layers" is presented heuristically without systematic validation. The paper does not explore how the choice of layer affects performance, nor does it justify why intermediate layers are better than later layers for this specific task. An ablation study varying the layer depth is essential to validate this key design choice and ensure the method's stability across different model architectures.

[1] ARGS: Alignment as Reward-Guided Search, ICLR 2024.

[2] SafeDecoding: Defending against Jailbreak Attacks via Safety-Aware Decoding, ACL 2024.

**Questions:**

1. Could you provide empirical evidence comparing the performance of SRR with directly ranking the final response or the final hidden state? This is critical to substantiate the claim that internal states capture more fine-grained safety signals.
2. Can you report comparative results on metrics like safety rate and benign task performance against more advanced baselines on the same datasets?
3. Can you compare the computational overhead of SRR with other inference-time aligning methods?
4. It seems that all LLMs are safety-aligned models. Does SRR still perform well for unaligned LLMs (e.g. uncensored models) as sampling refusal response from these models is more difficult?

---

### Meta-Review · Area_Chair_MogA · 2026-01-06

**Summary:**

All reviewers agree that the paper presents a simple and potentially deployable inference-time safety approach, which ranks multiple candidate generations using intermediate representations from a frozen LLM. However, there is broad consensus that the experimental evidence is insufficient to support the paper’s main claims and that the current evaluation does not convincingly demonstrate practical safety improvements over existing approaches. Specifically: (1) the baseline comparisons are weak and incomplete, relying mainly on a GPT-2 reward model and naïve generation strategies; (2) the evaluation focuses on ranking accuracy rather than generation-level safety outcomes such as attack success rate or harmful content rate; (3) key ablations are missing, particularly comparisons with ranking based on final-layer or text-level representations and a systematic analysis across layers; and (4) the method’s robustness in realistic settings is not sufficiently validated, with limited jailbreak testing and no assessment of over-refusal behavior.

**Reviewer Concerns:**

There is no rebuttal provided for this submission.

**Reviewer Scores:**

There is no rebuttal provided for this submission.

---

### Decision · Program_Chairs · 2026-01-26

Reject